# Tanapox Virus and Yaba Monkey Tumor Virus K3 Orthologs Inhibit Primate Protein Kinase R in a Species-Specific Fashion

**DOI:** 10.3390/v16071095

**Published:** 2024-07-08

**Authors:** Dewi Megawati, Jeannine N. Stroup, Chorong Park, Taylor Clarkson, Loubna Tazi, Greg Brennan, Stefan Rothenburg

**Affiliations:** 1Department of Medical Microbiology and Immunology, School of Medicine, University of California, Davis, CA 95616, USA; amegawati@ucdavis.edu (D.M.); jnstroup15@gmail.com (J.N.S.); chorong.park@northwestern.edu (C.P.); ltazi@ucdavis.edu (L.T.);; 2Department of Microbiology and Parasitology, Faculty of Medicine and Health Sciences, Warmadewa University, Denpasar 80239, Bali, Indonesia

**Keywords:** protein kinase R, poxviruses, K3L, yatapoxvirus, translational regulation

## Abstract

Yaba monkey tumor virus (YMTV) and Tanapox virus (TPV) are members of the Yatapoxvirus genus and can infect humans and other primates. Despite the threat posed by yatapoxviruses, the factors determining their host range are poorly understood. In this study, we analyzed the ability of YMTV and TPV orthologs of vaccinia virus K3 (called 012 in YMTV and TPV), which share 75% amino acid identity with one another, to inhibit PKR from 15 different primate species. We first used a luciferase-based reporter, and found that YMTV and TPV K3 orthologs inhibited PKR in a species-specific manner and showed distinct PKR inhibition profiles. TPV 012 inhibited PKR from 11 primates, including humans, substantially better than YMTV 012. In contrast, both K3 orthologs inhibited the other four primate PKRs comparably well. Using YMTV 012 and TPV 012 hybrids, we mapped the region responsible for the differential PKR inhibition to the C- terminus of the K3 orthologs. Next, we generated chimeric vaccinia virus strains to investigate whether TPV K3 and YMTV K3 orthologs could rescue the replication of a vaccinia virus strain that lacks PKR inhibitors K3L and E3L. Virus replication in primate-derived cells generally correlated with the patterns observed in the luciferase-based assay. Together, these observations demonstrate that yatapoxvirus K3 orthologs have distinct PKR inhibition profiles and inhibit PKR in a species-specific manner, which may contribute to the differential susceptibility of primate species to yatapoxvirus infections.

## 1. Introduction

Yatapoxviruses are a small group of *Chordopoxvirinae* that are pathogenic to primates, including humans. The Yatapoxvirus genus includes Yaba monkey tumor virus (YMTV) and Tanapox virus (TPV) [1]. Previously, Yaba-like disease virus (YLDV) was considered a third member; however, its genome shares 98.68% sequence identity with TPV, and can be considered a different variant of the same virus [2]. The genomes of YMTV and TPV share 75% sequence identity [3]. TPV was first isolated from human skin biopsies during a TPV outbreak in 1957 in Tana River Valley, Kenya [4,5]. YMTV was first isolated from an outbreak of subcutaneous tumors in monkeys in Yaba, Nigeria, in 1958 [6]. Yatapoxvirus genomes are A+T rich and range in size from 135 to 145 kb [2,3,7,8]. TPV and YMTV exhibit differences in disease presentations. TPV infection is characterized by vesicular skin lesions, whereas YMTV infection produces a localized histiocyte-filled tumor (histiocytomas) [9].

While both TPV and YMTV infect a wide range of primates, they appear to exhibit host specificity. Serological surveys have indicated that TPV is endemic in African and Malaysian monkeys, but not in Indian rhesus macaques or New World monkeys [4,5,10]. Human Tanapox disease is considered endemic to several regions of Africa and causes febrile illness and vesicular skin lesions similar to those produced in non-human primates. Sporadic cases have been identified in 30 locations spanning 6000 km from Sierra Leone to Tanzania, with larger outbreaks occurring from time to time [11]. Similarly, antibodies against YMTV have been found in great apes and Old World monkeys but not in New World monkeys or Indian rhesus macaques [10]. A serological survey of 456 primate sera including 26 chimpanzees, 326 Old World monkeys (African green monkeys, patas monkeys, baboons, colobus monkeys, and rhesus macaques), and 104 New World monkeys (spider monkeys, squirrel monkeys, owl monkeys, marmosets, and capuchin monkeys) indicated that antibodies against YMTV were evident in chimpanzees and Old World monkeys but not in any of the New World monkeys [10].

Following experimental subcutaneous inoculation with YMTV, tumor-like masses were detected in Asiatic rhesus macaques (*Macaca mulatta*, *Macaca irus*, and *Macaca speciosa*) but not in African green monkeys, mangabey monkeys, patas monkeys, mice, rats, rabbits, or guinea pigs [6,9,12], indicating host specificity. While no natural infection of YMTV in humans has been reported, the infection of human volunteers as well as an accidental infection of a laboratory worker have resulted in the development of mild histiocytomas [13]. Despite the threat posed to human health, the factors determining the host range of yatapoxviruses are poorly understood. In addition, vaccination with vaccinia virus does not appear to protect against yatapoxvirus infections [13,14]. Thus, it is important to study factors that determine the host range of yatapoxviruses.

A major determinant for the tropism of many viruses is the attachment and entry via host-specific cell receptors [15]. However, as poxviruses use ubiquitous cell surface receptors for entry, it is believed that the evasion of the innate immune response post entry is a more important determinant of poxvirus tropism [16,17]. Protein Kinase R (PKR) is a prominent host restriction factor against poxvirus infection (reviewed in [18]). PKR exists in a monomeric inactive form and is activated upon binding to double-stranded RNA (dsRNA) produced during infection by most viruses, which leads to PKR dimerization and autophosphorylation. Activated PKR phosphorylates the alpha subunit of eukaryotic initiation factor 2 (eIF2) [19]. Phosphorylated eIF2α has a high binding affinity for the regulatory core of the guanine nucleotide exchange factor eIF2B and prevents eIF2B from catalyzing GDP-GTP exchange to eIF2α [20]. As GTP-bound eIF2 is required for translation initiation, the resulting low availability of GTP-eIF2 leads to the shut-off of eIF2-dependent protein synthesis, including that of viral proteins. To overcome the antiviral effects of PKR activity, most poxviruses encode two PKR inhibitors, named E3 and K3 in vaccinia virus. E3 binds dsRNA and prevents PKR dimerization, whereas K3 inhibits PKR by acting as a pseudosubstrate for the eIF2α binding site [21,22,23,24].

It was previously shown that the inhibition of host PKR by K3 orthologs of several poxvirus families was a key determinant of host specificity [22,25,26,27,28]. Vaccinia virus lacking the E3L gene (VACVΔE3L) but maintaining the K3L gene was unable to replicate in HeLa cells but remained replication-competent in BHK21 cells, suggesting a role for K3 in determining host range [22]. This finding was supported by recent studies on leporipoxvirus, capripoxvirus, and orthopoxvirus K3 orthologs. The myxoma virus (MYXV) K3 ortholog M156 specifically inhibited rabbit PKR but failed to inhibit other PKR species [25]. Similarly, capripoxvirus K3 orthologs strongly inhibited human, goat, and sheep PKR, but only weakly inhibited mouse and cow PKR [26]. Our studies on the inhibition of PKR from a panel of mammalian species by orthopoxvirus K3 orthologs exhibited distinct inhibition profiles. Here, we used a luciferase-based assay and infections of mammalian cell lines to examine the ability of TPV and YMTV K3 orthologs to inhibit PKR derived from 15 primate species. Our results demonstrate that TPV and YMTV K3 orthologs have distinct PKR inhibition profiles and that they inhibit primate PKR in a species-specific manner.

## 2. Materials and Methods

### 2.1. Cell Lines

Tert immortalized gibbon fibroblasts (gibbon Tert-GF), HeLa cells (human, ATCC #CCL-2), HeLa PKR-knock-out (PKR^ko^) cells, and BSC-40 cells (ATCC CRL-2761) were kindly provided by Dr. Adam Geballe [29]. RK13 cells (rabbit) expressing E3 and K3 (designated RK13+E3L+K3L) were previously described [30]. The cells were grown in Dulbecco’s Modified Eagle’s Medium (DMEM, Life Technologies, Carlsbad, CA, USA) supplemented with 5% fetal bovine serum (FBS, Lonza Bioscience, Walkersville, MD, USA) or 10% FBS and 100 IU/mL penicillin/streptomycin (Gibco, Life Technologies, Carlsbad, CA, USA). The RK13+E3L+K3L cell culture medium contained 500 μg/mL geneticin and 300 μg/mL zeocin (Life Technologies).

### 2.2. Plasmids

A library of primate PKR expressing plasmids were kindly provided by Dr. Nels Elde [31]. Primate PKR and viral antagonist genes VACV K3L (YP_232916.1), TPV 012 (EF420156.1), YMTV 012 (AY386371.1) were subcloned into the pSG5 expression vector for luciferase-based reporter assays. pGL3 luciferase reporter vector was purchased from Promega, Madison, WI, USA. Cloning was carried out using Gibson assembly techniques (New England Biolabs, Ipswich, MA, USA) [32]. To monitor VACV K3L, TPV 012, and YMTV 012 gene expressions, these genes were cloned into C-terminus DYK-tagged pSG5 vector for transient transfection assay. To generate hybrid _N_TPV-_C_YMTV and _N_YMTV-_C_TPV K3 orthologs, we PCR-amplified _N_YMTV 012 (nucleotide 1–99), _C_YMTV 012 (nucleotide 100–267), _N_TPV 012 (nucleotide 1–99), and _C_TPV 012 (nucleotide 100–267) and cloned the TPV 012 and YMTV 012 hybrid genes into pSG5 vector using Gibson assembly. To generate recombinant VACV expressing TPV 012 and YMTV 012 genes, the yatapoxvirus K3 ortholog genes were cloned into p837-GOI-mCherry-E3L, as previously described [33].

### 2.3. Luciferase-Based Reporter Assays

Luciferase-based assays were performed as previously described [34]. Briefly, 5 × 10^4^ HeLa PKR^ko^ cells were seeded per well in 24-well plates overnight. The HeLa PKR^ko^ cells were co-transfected with 200 ng of the indicated PKR expression vector, 200 ng of each viral antagonist expression vector, and 50 ng of pGL3 firefly luciferase expression vector (Promega), following the manufacturer’s protocol. Cells were lysed with mammalian lysis buffer (GE Healthcare, Chicago, IL, USA) at 48 h post transfection. Luciferase activity was measured using a GloMax luminometer (Promega) by adding luciferin (Promega) reagent to the cell lysates, as per the manufacturer’s recommendations. Data are presented as relative luciferase activity in which all data were normalized to pSG5 empty vector. Experiments were conducted in triplicate for each of the three independent experiments.

### 2.4. Virus and Infections

Vaccinia virus variant vP872 [35] was kindly provided by Dr. Bertram Jacobs. VC-R4+VACV K3L and VC-R4+sheeppox 011 (SPPV 011), as well as the generation of VC-R4 from vP872 variant, were previously described [26,33]. The generation of chimeric vaccinia virus, VC-R4+TPV 012, and VC-R4+YMTV 012, was carried out through the scarless integration of the open reading frames of the yatapoxvirus K3L orthologs into the E3L locus [33]. The chimeric viruses were plaque-purified twice, and the K3L ortholog gene integrations were confirmed by means of Sanger sequencing. The viruses were purified by means of zonal sucrose gradient centrifugation, and the virus titer was determined on confluent 12-well plates of RK13+E3L+K3L cells. Plaque assays were performed with confluent 6-well plates of the indicated cell lines, which were infected with 50 plaque-forming units (pfu) of each indicated virus. One hour post infection, the media were replaced with DMEM containing 5% FBS, 1% carboxymethylcellulose (CMC), and 100 U P/S. After 48 h, cells were stained with 0.1% crystal violet, and the excess staining was washed with water. The plates were imaged using an iBright Imaging System (Invitrogen, Waltham, MA, USA). Virus infection assays were performed in confluent 6-well plates of the indicated cells. The cells were infected with each indicated virus at a MOI of 0.01 to assess replication in multiple-cycle replication assays. Cells and supernatants were collected at 30 h post infection and subjected to three rounds of freezing at −80 °C and thawing at 37 °C. Lysates were sonicated twice for 15 s, at 50% amplitude (Qsonica Q500, Newtown, CT, USA). Virus titers were measured in plaque-forming units per mL (pfu/mL) on RK13+E3L+K3L cells. Virus titer data were analyzed using one-way ANOVA followed by Tukey’s multiple comparisons test (GraphPad Prism version 10.2.3).

### 2.5. PCR of Viral Genomic DNA

HeLa PKR^ko^ cells were seeded in 10 cm dishes to a confluency of 90–100%. The cells were then infected with the indicated viruses at a MOI of 0.1 for 24 h. Viral genomic DNA (gDNA) extraction was performed as previously described [36]. About 100 ng of the isolated viral gDNA was used as a template in a PCR targeting K3L ortholog genes using Phusion High Fidelity DNA polymerase (NEB #M0530L). The forward primer sequence was 5′ GACGAACCACCAGAGGATGATG 3′ and the reverse primer sequence was 5′ AGTACTACAATTTGTATTTTTTAATCTATCTCA 3′. PCR products were gel-purified with Monarch DNA Gel Extraction Kit (NEB #T1020) and Sanger sequencing was performed to confirm the correct integration of the K3L orthologs.

### 2.6. Immunoblot Analyses

To determine the level of K3 ortholog expression in a transient transfection system, 4 × 10^5^ HeLa PKR^ko^ cells were seeded in a 6-well plate. On the next day, the cells were transfected with 1 µg of pSG5 empty vector, DYK-tagged VACV K3L, DYK-tagged TPV 012, and DYK-tagged YMTV 012, using a Genjet to DNA ratio of 2:1. At 48 h post transfection, cells were washed with PBS, lysed with 1% sodium dodecyl sulfate (SDS) in DPBS, and sonicated at 50% amplitude for 5 s twice. About 10 µg of proteins were run on 12% SDS polyacrylamide gels (Biorad, Hercules, CA, USA) and transferred to polyvinyl difluoride (PVDF, GE Healthcare) membranes. The membranes were blocked with SuperBlock blocking buffer (Thermofisher, Waltham, MA, USA) for 1 h and probed with primary antibodies against Flag M2 (Sigma F1804, St. Louis, MO, USA) and beta-actin (Sigma A1978) at a dilution of 1:2000 in SuperBlock blocking buffer (Thermofisher) for 1 h at room temperature. After washing with Tris Buffered Saline 0.1% Tween® 20 (TBST, Fisher Bioreagents), the membranes were probed with secondary antibody horseradish peroxidase-conjugated donkey anti-mouse (Jackson ImmunoResearch Laboratories Inc., 715-035-150, West Grove, PA, USA) at a dilution of 1:10,000 in 1% (*w*/*v*) non-fat milk dissolved in TBST for 1 h. The membranes were washed three times for 10 min, and proteins were detected with Amersham™ ECL™ (GE Healthcare). Images were taken using the iBright Imaging System (Invitrogen).

To determine the level of K3L ortholog expression by the chimeric vaccinia virus, 8 × 10^5^ HeLa PKR^ko^ cells were seeded in a 6-well plate. On the next day, the cells were infected with VC-R4, VC-R4+VACV K3L, VC-R4+TPV 012, and VC-R4+YMTV 012 at a MOI of 3 to ensure that the vast majority of cells were infected. Protein lysates were collected at 24 h post infection. A total of 12 µg of protein was electrophoresed using 12% SDS polyacrylamide gels and transferred to polyvinyl difluoride (PVDF, GE Healthcare) membranes. The membranes were probed with primary antibodies for beta-actin (Sigma A1978) at a dilution of 1:2000 in Thermofisher Superblock blocking buffer, primary antibodies against TPV 012 (Genscript, Piscataway, NJ, USA) at a dilution of 1:1000 in 5% milk dissolved in TBST, and VACV-K3 at a dilution of 1:500 in 5% milk dissolved in TBST overnight at 4 °C. Anti-TPV 012 (cVQVIRTDKLKGYVDVRHIT) and anti-VACV K3 (cKVIRVDYTKGYIDVNYKRM) were custom-produced using peptide–KLH conjugate in New Zealand rabbit (GenScript). After being washed with TBST three times, the membranes were probed with horseradish peroxidase-conjugated donkey anti-rabbit IgG (H+L) secondary antibody (Invitrogen, A16023) at a dilution of 1:10,000 in 1% milk for anti-TPV 012 and anti-VACV K3, and with horseradish peroxidase-conjugated donkey anti-mouse IgG (H+L) (Jackson ImmunoResearch Laboratories Inc., 715-035-150) at a 1:10,000 dilution in TBST containing 1% (*w*/*v*) non-fat milk for beta-actin. The membranes were washed three times for 10 min, and proteins were detected with Amersham™ ECL™ (GE Healthcare). Images were taken using the iBright Imaging System (Invitrogen).

### 2.7. Phylogenetic Analysis

The amino-acid sequences of PKR from 21 different primate species (Table 1) were used to generate the phylogenetic tree. The multiple sequence alignment was performed using MUSCLE [37], and the resulting alignment was used to construct a phylogenetic tree using the maximum likelihood approach, as implemented in the program PhyML 3.0 [38]. The nodal support of the phylogenetic tree was assessed by means of a bootstrap analysis of 100 replicates. The resulting phylogenetic tree was visualized using FigTree v1.4.4 (available at: http://tree.bio.ed.ac.uk/software/figtree/).

## 3. Results

### 3.1. Species-Specific Inhibition of Primate PKRs by Yatapoxvirus K3 Orthologs

The inhibition of PKR by K3 orthologs has been shown to be a key determinant for the host range of poxviruses [24,25,26,27,28,34]. In this study, we examined the role of the K3 orthologs TPV 012 and YMTV 012 in inhibiting PKR derived from a diverse set of primate species. The comparison of the amino acid sequences between TPV and YMTV 012 revealed 22 amino acid differences (75% sequence identity). Yatapoxvirus K3 orthologs and VACV K3 showed about 36% sequence identity (Figure 1A,B). We cloned these three orthologs into the pSG5 expression vector with two C-terminal FLAG tags and transfected them into HeLa PKR^ko^ cells. The Western blot analysis of whole cell lysates showed that vaccinia K3 and TPV 012 were expressed at comparable levels, whereas YMTV 012 was expressed at higher level than the former (Figure 1C). We also raised polyclonal antibodies against a conserved epitope of TPV 012 (Figure 1A). Western blot HeLa PKR^ko^ cell lysates transiently transfected with untagged K3L orthologs showed a higher expression of YMTV 012 (Figure 1D). A previously generated antibody raised against VACV K3 detected both VACV K3 and YMTV 012 (Figure 1D), which could be explained by shared epitopes.

We next tested the sensitivities of 15 different primate species to VACV K3 and the yatapoxvirus K3 orthologs in an established luciferase-based expression assay in which PKR-deficient HeLa cells are co-transfected with a luciferase reporter plasmid, PKR, and PKR inhibitors [25,34]. In this assay, PKR is activated by dsRNA that is formed by overlapping transcripts generated from the transfected plasmids [40] and suppresses the translation of reporter mRNA, which serves as a proxy to quantify relative translation. Luciferase activity is thus inversely correlated with PKR activity, and the inhibition of PKR leads to elevated luciferase activity. We first co-transfected cells with PKRs from 15 primate species representing apes (human, gorilla, chimpanzee, orangutan, and white-cheeked gibbon), Old World monkeys (colobus monkey, François’ leaf monkey, baboon, sooty mangabey, rhesus macaques, talapoin monkey, African green monkey, and patas monkey) and New World monkeys (tamarin and dusky titi), together with the luciferase reporter plasmid to assess PKR activity. All PKRs showed a strong and comparable inhibition of luciferase activity (Figure 2A). We next co-transfected each PKR with either K3 ortholog and the luciferase reporter plasmid (Figure 2B). Most primate PKRs tested were largely resistant to VACV K3 (less than 2-fold inhibition). Orangutan and white-cheeked gibbon PKR were weakly inhibited (about 2-fold), while rhesus macaque and tamarin PKR were inhibited at intermediate low levels (about 4-fold). In total, 11 of the 15 primate PKRs were inhibited substantially better by TPV 012 than by YMTV 012. PKRs from chimpanzee, white-cheeked gibbon, colobus monkey, and François’ leaf monkey were inhibited comparably well by TPV 012 and YMTV 012. In addition to different inhibition profiles when comparing TPV 012 with YMTV 012, species-specific differences in PKR susceptibility were observed, with orangutan, François leaf monkey, and rhesus macaque PKR being most strongly inhibited, and nine other PKRs being intermediately inhibited by TPV 012. In contrast, nine tested PKRs were only weakly inhibited by YMTV 012.

We also projected the relative sensitivities of the tested PKRs to the K3 orthologs on a phylogenetic tree generated with PKR amino acid sequences from a panel of 21 primate species using rabbit PKR as an outgroup to visualize the relatedness among the primate species PKRs (Figure 3). This tree recapitulates the relatedness of primates PKRs from an earlier study. The closer relatedness of human PKR to gorilla PKR than to chimpanzee PKR, which is different from the generally accepted primate phylogeny, likely is a result of positive selection during PKR evolution [31]. We used a previously described scale to grade relative PKR inhibition [28], with 1 for no or very weak inhibition (<2-fold inhibition), 2 for weak inhibition (2- to 3-fold inhibition), 3 for intermediate low inhibition (3- to 5-fold inhibition), 4 for intermediate high inhibition (5- to 7-fold inhibition), 5 for high inhibition (7- to 10-fold inhibition), and 6 for very high inhibition (>10-fold inhibition).

### 3.2. Differential PKR Inhibition Was Governed by the C-Terminus of Yatapoxvirus K3 Orthologs

The amino acid sequence alignment of YMTV and TPV K3 orthologs identified 22 amino acid differences between the two orthologs, which are distributed throughout the protein sequence. To investigate which regions are important for the differential PKR inhibition by yatapoxvirus K3 orthologs, we generated constructs encoding hybrid YMTV-TPV K3 orthologs in which the C-terminal regions of the two inhibitors were swapped. We took advantage of the shared region in the middle part of the gene (FTVFLPEFG) to separate the N-and C-termini, which leaves 11 amino acid differences in each terminus (Figure 4A). We analyzed the ability of these hybrid proteins to inhibit human, colobus monkey, and dusky titi PKR using the luciferase assay. The hybrid _N_TPV-_C_YMTV exhibited inhibition profiles that were more similar to YMTV 012 for each PKR tested (Figure 4B). Correspondingly, the _N_YMTV-_C_TPV hybrid showed a similar inhibition pattern to wild-type TPV 012. These results indicate that the C-terminus of TPV 012 and YMTV 012 is important for the observed differential PKR inhibition, although we cannot exclude a contribution of the N-terminus.

### 3.3. Chimeric Viruses Expressing TPV 012 or YMTV 012 Displayed Cell-Type-Specific Differences in Plaque Formation and Virus Replication

Next, we investigated whether the ability of TPV 012 and YMTV 012 orthologs to inhibit primate PKRs correlated with the ability of the viral genes to rescue the replication of a VACV strain that lacks PKR inhibitors in primate-derived cell lines. The VACV strain VC-R4 (VACVΔE3LΔK3L) is a highly attenuated vaccinia virus variant that can only replicate in PKR-deficient cell lines or cell lines expressing PKR antagonists [33]. We inserted either TPV 012 or YMTV 012 into VC-R4 at the E3L locus using the scarless integration method as previously described, designating the chimeric viruses VC-R4-TPV 012 or VC-R4-YMTV 012, respectively [33]. To analyze the expression of the K3 orthologs by these chimeric viruses, we infected HeLa PKR^ko^ cells and performed Western blots using custom-made antibodies at 24 h post infection. In agreement with the transfection data (Figure 1), YMTV 012 was detected at a higher level compared to TPV 012 (Figure 5A). We next examined the plaque formation of these viruses in comparison to VACV vP872 (which lacks K3L but contains the second VACV PKR inhibitor E3), VC-R4+VACV K3L, and VC-R4+SPPV 011 which expresses the sheeppox virus K3 ortholog [26], in different primate-derived cell lines: HeLa (human), gibbon Tert-GF (gibbon), and BSC-40 and PRO1190 (African green monkey). As control, we used RK13+K3L+E3L cells (rabbit), which express VACV E3 and K3 and in which VC-R4 is permissive. We infected these cells with 50 PFU/well. As expected, all viruses formed plaques in permissive RK13+K3L+E3L cells. VC-R4 only formed plaques in RK13+K3L+E3L cells, and vP872 and VC-R4+SPPV 011 formed plaques in all cells tested (Figure 5B).

VC-R4+VACV K3L developed small plaques only in BSC-40 cells. VC-R4+TPV 012 infection led to the formation of plaques in all tested cells with comparable sizes to plaques formed by both VC-R4+SPPV 011 and vP872. Relative to VC-R4+TPV 012, VC-R4+YMTV 012 formed smaller plaques in HeLa and gibbon Tert-GF cells and failed to cause plaque formation in both African green monkey cells. In general, based on the plaque assay, TPV 012 rescued VC-R4 replication substantially better than YMTV 012 in primate-derived cell lines, despite the higher expression of YMTV 012.

We next measured virus titers to determine the capacity of yatapoxvirus K3 orthologs to restore the replication of VC-R4. We infected RK13+K3L+E3L and primate-derived cell lines HeLa (human), gibbon Tert-GF, BSC-40 (African green monkeys), and PRO1190 (African green monkeys) cells with either VC-R4, vP872, VC-R4+VACV K3L, VC-R4+SPPV 11, VC-R4+TPV 012, or VC-R4+YMTV 012 at a MOI of 0.01. Cell lysates were collected at 30 h post infection, and the virus titer was determined in RK13+K3L+E3L cells. Consistent with the plaque assays and luciferase assays, all viruses replicated to comparable titers in RK13+K3L+E3L cells (Figure 6). VC-R4 did not replicate in any cells tested (virus titer below 103 pfu/mL), except in RK13+K3L+E3L cells. VACV K3L rescued VC-R4 virus replication in all cells tested to variable extents, with virus titers about 10- to 100-fold lower than vP872. In HeLa and gibbon Tert-GF cells, SPPV 011 and TPV 012 both restored the replication of VC-R4 to a level comparable to vP872. In BSC-40 and PRO1190 cells, TPV 012 and SPPV 011 rescued VC-R4 replication comparably. Consistent with the plaque assays, VC-R4+YMTV 012 replicated to slightly lower levels than VC-R4+TPV 012 in HeLa cells, although this difference was not significant. In contrast, VC-R4+YMTV 012 significantly replicated with less efficiency than VC-R4+TPV 012 in gibbon Tert-GF cells (11-fold lower), BSC-40 cells (1394-fold lower), and PRO1190 cells (121-fold lower).

## 4. Discussion

Vaccinia virus K3 is a poxvirus host range factor that acts by inhibiting the host restriction factor PKR. Homology between the K3 and the S1 domain of eIF2α allows K3 to function as a pseudosubstrate of PKR, competitively inhibiting eIF2α phosphorylation [19]. This interaction between PKR and K3 has been studied using yeast and mammalian cell assay systems [41,42]. Previously, Elde et al. have studied the interaction of 10 primate PKR variants with VACV K3 and showed that PKR from Old World monkeys and New World monkeys were generally susceptible to VACV K3, whereas PKR from hominoids were resistant to VACV K3, indicating a species-specific inhibition of primate PKR by VACV K3 [31]. We recently described that K3 orthologs from myxoma virus, which infects rabbits and hares, and capripoxviruses, which infect sheep, goats, and cattle, also inhibit host PKRs in a species-specific manner [25,26,43]. Our recent studies revealed that K3 inhibition could not be predicted by the phylogenetic relatedness of PKR alone [28]. Therefore, in this study, we experimentally determined whether K3 orthologs of the Yatapoxvirus genus also exhibit host-specific PKR inhibition.

Yatapoxviruses infect primates and can cause zoonotic infections in humans. Here, we analyzed whether TPV and YMTV K3 orthologs inhibit PKR from 15 different primate species in comparison to VACV K3. Our results confirm and extend the results of species-specific VACV K3 inhibition previously reported in both yeast-based [31] and luciferase-based assays [28]. White-cheeked gibbon PKR was largely resistant to VACV K3, whereas VACV K3 strongly inhibited tamarin PKR and rhesus macaque PKR. Interestingly, dusky titi PKR, which was previously shown to be susceptible to VACV K3 in the yeast-based assay, appeared to be largely resistant to VACV K3 in our luciferase-based assays. Although the yeast assay can be used to predict the inhibition of PKR by viral antagonists, yeast expression systems have very different post-translational modifications relative to mammalian cells, which might affect protein function [44]. In general, our data showed that TPV 012 and YMTV 012 exhibited distinct PKR inhibition profiles and that these K3 orthologs inhibited primate PKR in a species-specific manner. All tested primate PKRs were sensitive to TPV 012, although to different degrees. We found a distinct profile of inhibition by YMTV 012 within the Old World monkeys in which talapoin monkey, African green monkey, and patas monkey PKRs were largely resistant to YMTV 012 inhibition. In contrast, rhesus macaque PKR was largely sensitive, supporting that the interaction between K3 orthologs and PKR could only be partially inferred by phylogenetic relatedness. This finding is in line with subcutaneous inoculation studies, showing that YMTV induced tumor formation in rhesus monkeys, whereas no tumors were observed in sooty mangabey, patas monkey, and African green monkeys [6,9,12].

TPV 012 and YMTV 012 differ by 22 amino acids out of 88 residues (Figure 1). K3 orthologs from the orthopoxvirus genus share a higher sequence identity (80.7–100%) among their members [28] than TPV 012 and YMTV 012. We found that residues within the C-terminus region are important for the differential PKR inhibition by these orthologs. There are two critical conserved motifs in the C-terminus of the K3 orthologs: the PKR recognition motif and helix insert region [19]. The PKR recognition motif, which includes residues K74, Y76, and D78, provides important determinants for high-affinity binding to PKR [19]. TPV 012 and YMTV 012 orthologs have identical residues in the PKR recognition motif “KGYVD”, one amino acid difference compared to the PKR recognition motif “KGYID” of VACV K3. The helix insert of eIF2α appears to undergo conformational change upon binding to PKR, resulting in the Ser51 site of eIF2α becoming fully accessible to the phosphoacceptor binding site of PKR [45]. The K3 helix insert region is structurally homologous to the region adjacent to Ser51 in eIF2α. The unique conformation of the helix insert region of K3L functions as an inhibitor of PKR activation by preventing PKR trans-autophosphorylation and acting as a pseudosubstrate of PKR [45]. In addition, we and others have shown that one or more amino acid residues in orthopoxvirus K3 orthologs are responsible for the differential PKR inhibition and that the amino acid variants are predominantly located in the helix insert region [27,28].

In this study, the function of yatapoxvirus K3 proteins was further examined in the context of virus infection. The E3L and K3L double-knockout vaccinia virus strain VC-R4 (VACVΔE3LΔK3L) can only replicate in PKR-deficient cell lines or cell lines that express PKR antagonists. By inserting K3L orthologs from different poxviruses in the E3L locus of VC-R4, we can analyze the chimeric viruses’ ability to replicate in PKR-competent cell lines. The advantage of using the chimeric virus system is that it allows us to assess the ability of K3L orthologs in rescuing the virus in PKR-competent cells. As expected, VC-R4 could not replicate in all tested cells, except RK13+E3L+K3L cells. VC-R4 formed smaller plaques than vP872, which can be explained by incomplete PKR inhibition by the ectopically expressed E3 and K3, and is consistent with the slightly less efficient replication of VC-R4 than vP872 in this cell line [46]. VC-R4 expressing SPPV 011 and TPV 012 replicated as efficiently as vP872 in all tested cell lines, as shown in plaque assays and virus titer. In contrast, VC-R4+VACV K3L and VC-R4+YMTV 012 formed smaller plaques in all tested cell lines, except in RK13+E3L+K3L where these viruses formed comparable plaque size with other viruses. Together these results indicate that VACV K3L and YMTV 012 are generally weaker inhibitors for the PKR orthologs that we tested. This result was recapitulated in the virus replication assay. There was an 11-fold higher virus titer in VC-R4+TPV 012-infected gibbon Tert-GF cells than in VC-R4+YMTV 012-infected cells. The difference in virus titer was more pronounced in PRO1190 and BSC-40 cells, with about 100- to 1000-fold lower virus titer in VC-R4+YMTV 012 infections. Although both cell lines are derived from African green monkeys, BSC-40 are immortalized epithelial cells and PRO1190 are primary fibroblasts. Differential susceptibility to various VACV recombinants by these two cell lines is well established, and may in part be due to three non-synonymous variants in PRO1190 PKR relative to BSC-40 PKR, although other host proteins, e.g., RNase L, have been implicated in some studies, suggesting that this phenotype is likely multifactorial [47,48,49]. While the general patterns observed in the luciferase-based assay were recapitulated in the infection assay, we observed some differences in the magnitude of inhibition.

Overall, the data presented here support previous observations regarding species-specific PKR inhibition by VACV K3 orthologs and extend them to a new genus of poxviruses [27,28,31]. Within primate species PKR, TPV 012 exhibited a broader range of PKR inhibition than did YMTV 012 and VACV K3. VACV K3 appeared to have a narrow range of PKR inhibition within primate species but efficiently inhibited PKR from a wide range mammals PKR [28]. It is worth mentioning that the ability of a virus to replicate and disseminate efficiently in its host is dependent on the entire interactome between the host and the virus [50]. In addition to K3L, yatapoxviruses encode several host range genes, including E3L, C7L, M11L, Serpins, P28-like, and B5R [51]. The role of the E3L gene of YMTV and C7L of YLDV in host range factors has been previously described [52,53]. However, the role of other genes as host range factors needs to be experimentally confirmed.

## 5. Conclusions

The possible reemergence of smallpox-like diseases through the zoonosis of animal poxviruses is a major public health concern. Members of the Yatapoxvirus genus infect humans and other primates and have caused outbreaks throughout Africa for the past 60 years. In addition, human infections with yatapoxviruses have been reported in animal handlers at primate centers in the United States and among travelers who visited Africa. Despite the threat posed to human health, the factors determining the host range of these viruses remain poorly understood. Here, we describe the inhibitory profiles of TPV and YMTV K3 orthologs against primate PKRs, and show that they display virus and host species-specific PKR inhibition. The TPV and YMTV K3 orthologs exhibited distinct PKR inhibition profiles, with TPV 012 generally inhibiting PKRs more strongly than YMTV 012. The species-specific PKR inhibition by the TPV and YMTV K3 orthologs may contribute to the differential susceptibility of primate species to TPV and YMTV infections.

## Figures and Tables

**Figure 1 viruses-16-01095-f001:**
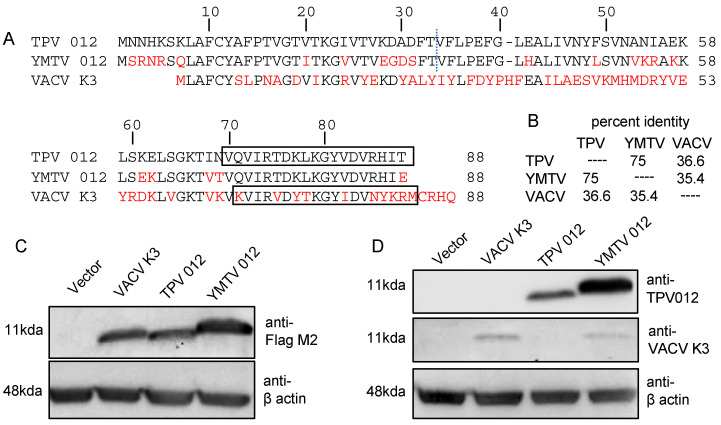
Amino acid differences among TPV 012, YMTV 012, and VACV K3. (**A**). Residues differing from TPV 012 are highlighted in red. The regions to the left and right of the dotted line were considered the N-terminal and C-terminal regions of the inhibitors, respectively. Sequences used to generate antibodies against TPV 012 and VACV K3 are boxed. (**B**). Percent identities between tested K3 orthologs were calculated from the multiple sequence alignment using Clustal Omega [39]. (**C**,**D**). HeLa PKR^ko^ cells were transfected with 1 µg of either pSG5 empty vector, pSG5 encoding FLAG-tag VACV K3, FLAG-tag TPV 012, or FLAG-tag YMTV 012. Proteins were harvested at 48 h post transfection, and protein expressions were detected with anti-Flag M2, anti-TPV 012, anti-VACV K3, and anti-β actin antibodies as indicated.

**Figure 2 viruses-16-01095-f002:**
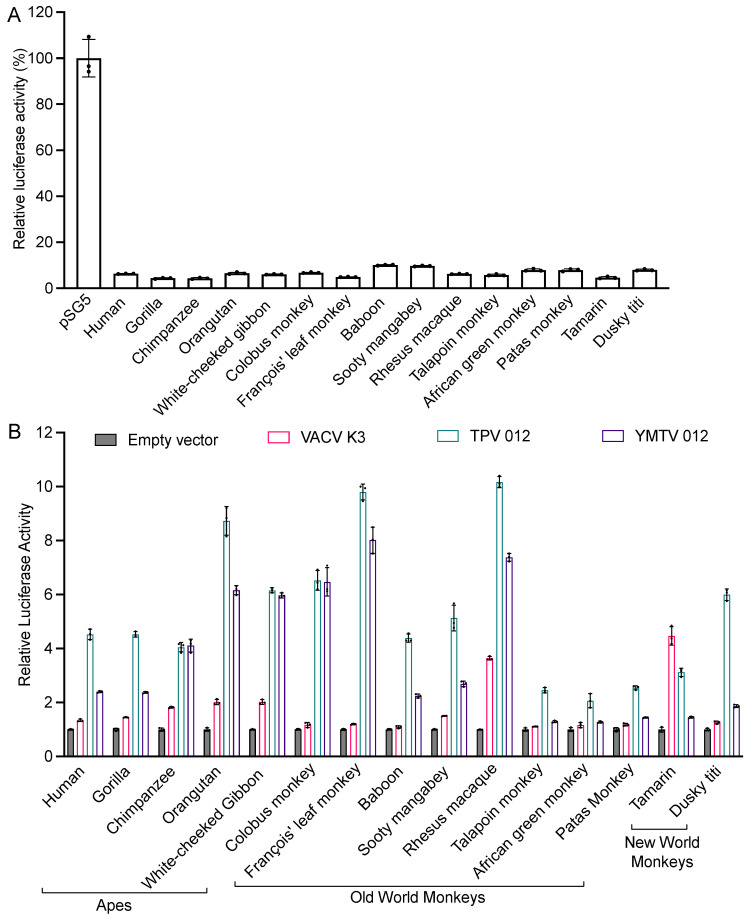
Differential sensitivities of tested PKRs to VACV K3, TPV 012, and YMTV 012 orthologs. (**A**). HeLa PKR^ko^ cells were co-transfected with expression vectors encoding firefly luciferase (50 ng) and with either empty vector or PKR (200 ng) from the indicated species. Luciferase activities were measured 48 h after transfection and normalized to empty vector transfected cells to obtain relative luciferase activities. (**B**). HeLa PKR^ko^ cells were transfected with expression vectors encoding firefly luciferase (50 ng), PKR (200 ng), and PKR inhibitor (200 ng) from the indicated species. Luciferase activities were measured 48 h after transfection and normalized to PKR-only transfected cells to obtain relative luciferase activities. Error bars represent the standard deviations from three independent transfections. The luciferase assay results shown are representative of at least three independent experiments.

**Figure 3 viruses-16-01095-f003:**
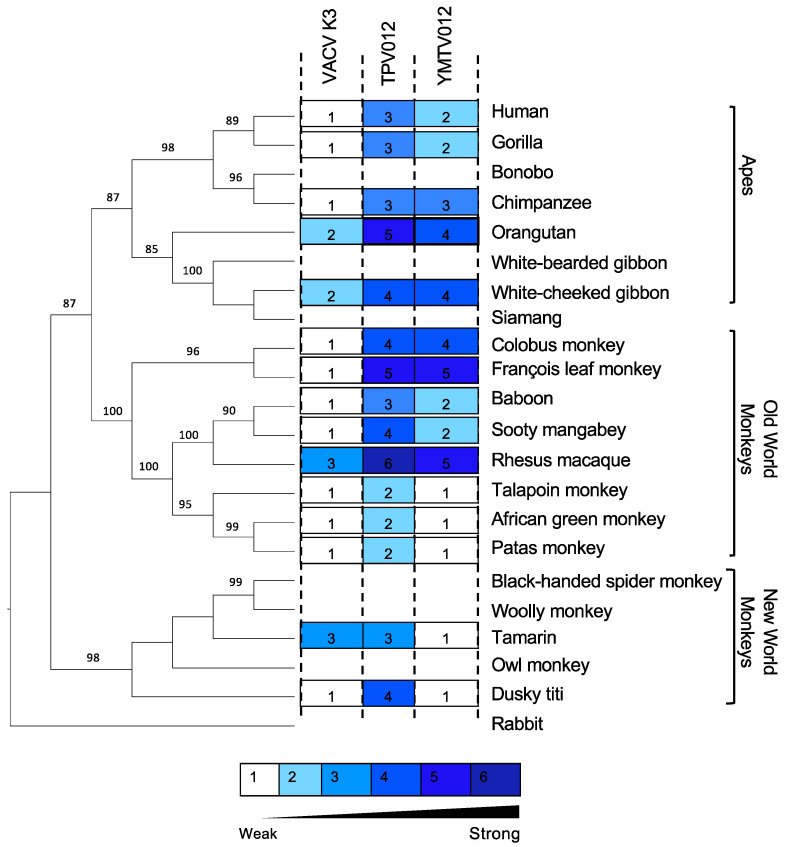
Sensitivities of tested PKRs projected on a PKR phylogenetic tree. We generated a phylogenetic tree with PKR protein sequences from a panel of 21 primate species and used rabbit PKR as an outgroup. The tree was built by PhyML with 100 bootstrap replicates and based on the maximum-likelihood method. Relative sensitivities of PKRs to either VACV K3, TPV 012, or YMTV 012 from multiple luciferase-based reporter assays are shown on a scale from 1 to 6. Scale 1 = no or very weak inhibition (≤2-fold increase in luciferase readout); 2 = weak inhibition (2- to 3-fold); 3 = intermediate low inhibition (3- to 5-fold); 4 = intermediate high inhibition (5- to 7-fold); 5 = high inhibition (7- to 10-fold); 6=very high inhibition (≥10-fold).

**Figure 4 viruses-16-01095-f004:**
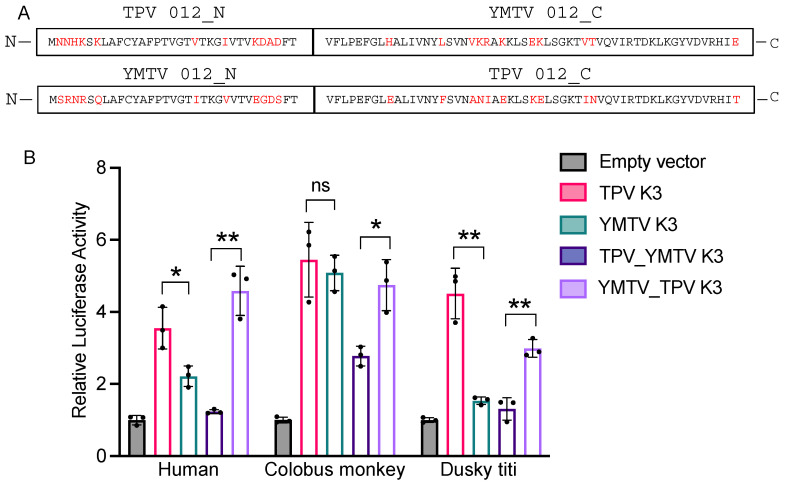
Identification of region of TPV 012 and YMTV 012 orthologs that confer differential PKR inhibition. (**A**). The predicted amino acid sequences of TPV 012 and YMTV 012 hybrid proteins. Constructs were designed such that the C-terminal regions of the two inhibitors were swapped to generate hybrid proteins. Different residues among K3 orthologs are highlighted in red. (**B**). Human HeLa-PKR^ko^ cells were transfected with expression vector encoding firefly luciferase, indicated PKRs, and either hybrid _N_TPV-_C_YMTV or _N_YMTV-_C_TPV inhibitors. The inhibition of indicated PKRs by TPV 012 and YMTV 012 serves as control. Luciferase activity was measured 48 h after transfection. Luciferase light units were normalized to PKR-only transfected cells to obtain relative luciferase activities. Error bars represent triplicate transfections and data are representative of three independent experiments. The data were analyzed using the unpaired *t*-test (GraphPad). *p*-values are indicated as follows: ns (non-significant) *p* > 0.05, * *p* < 0.05, ** *p* < 0.01.

**Figure 5 viruses-16-01095-f005:**
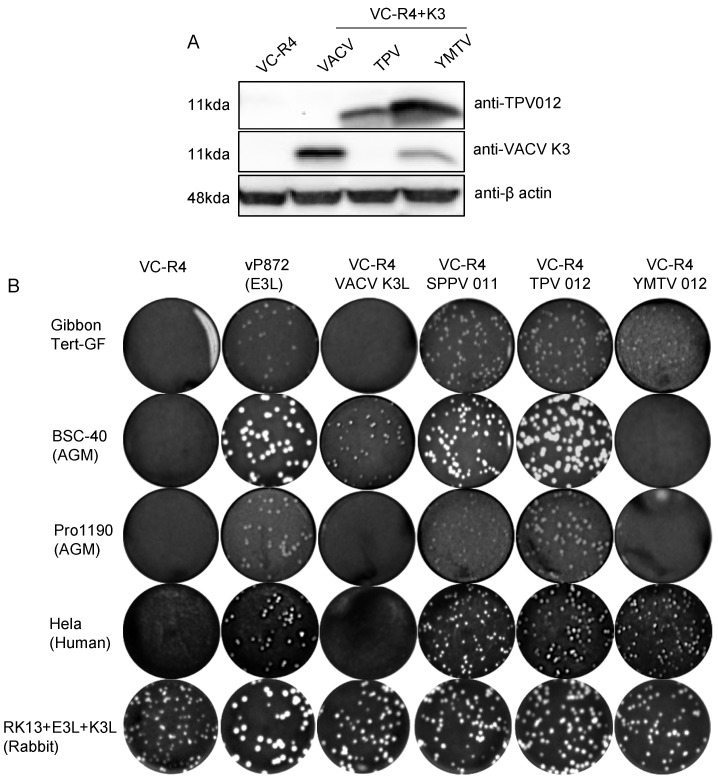
Generation of chimeric VC-R4 expressing yatapoxvirus K3 orthologs. (**A**). HeLa PKR^ko^ were infected with VC-R4, VC-R4+VACV K3, VC-R4+SPPV 011, VC-R4+TPV 012, or VC-R4+YMTV 012. Protein lysates were collected at 24 h post infection. The expression of K3 orthologs was determined by immunoblotting with the indicated antibodies. (**B**). Plaque formation of VC-R4 expressing K3 orthologs. A variety of primate-derived cells and RK13+E3L+K3L were infected with 50 pfu/mL of the indicated viruses, and plaque formation was visualized with crystal violet at 48 h after infection.

**Figure 6 viruses-16-01095-f006:**
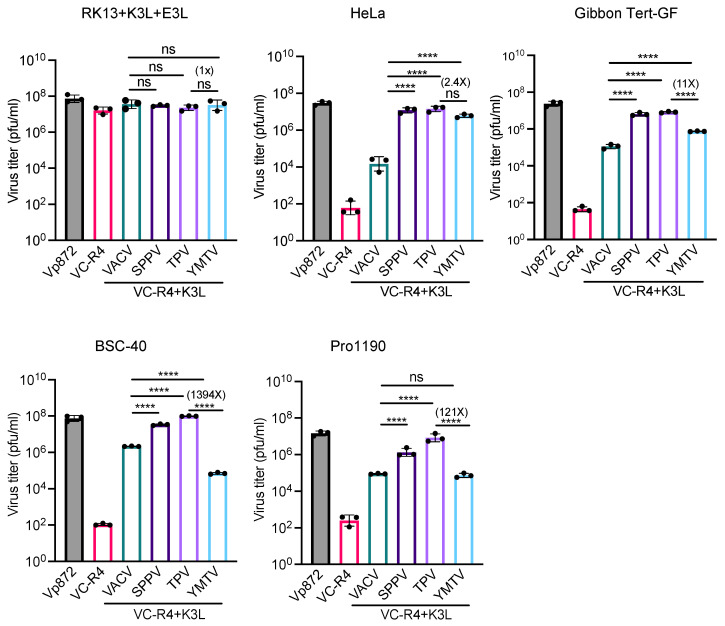
Virus titer of chimeric vaccinia viruses in primate-derived cell lines. A variety of primate-derived cells and RK13+E3L+K3L were infected with indicated viruses at a MOI of 0.01. Cell lysates were harvested at 30 h post infection and virus titers were determined in RK13+E3L+K3L cells. Error bars represent the standard deviations from three independent infections. Fold changes in virus titer between VC-R4+TPV 012 and VC-R4+YMTV 012 are shown. Data were analyzed using one-way ANOVA followed by Tukey’s multiple comparisons test. *p*-values are marked as follows: ns = *p* > 0.05, **** *p* < 0.0001.

**Table 1 viruses-16-01095-t001:** Accession numbers of genes used for the phylogenetic analysis.

Common Name	Scientific Name	GenBank Accession
Human	*Homo sapiens*	NM_002759
Gorilla	*Gorilla gorilla*	EU733258.1
Bonobo	*Pan paniscus*	EU733255.1
Chimpanzee	*Pan troglodytes*	EU733256.1
Orangutan	*Pongo pygmaeus pgmaeus*	EU733259.1
White-cheeked gibbon	*Nomascus leucogenys*	EU733257.1
White-bearded gibbon	*Hylobates albibarbis*	EU733270.1
Siamang	*Symphalangus syndactylus*	EU733271.1
Colobus monkey	*Colobus guereza*	EU733267.1
François’ leaf monkey	*Trachypithecus francoisi*	EU733268.1 *
Baboon	*Papio anubis*	XM_009184002.4 *
Sooty mangabey	*Cercocebus torquatus atys*	EU733262.1
Rhesus macaque	*Macaca mulatta*	EU733261.1
Talapoin monkey	*Miopithecus talapoin talapoin*	EU733269.1
African green monkey	*Clorocebus aethiops*	EU733254.1
Patas monkey	*Erythrocebus patas*	EU733260.1
Black-handed spider monkey	*Ateles geoffroyi*	EU733263.1
Woolly monkey	*Lagothrix lagotricha*	EU733266.1
Tamarin	*Saguinus labiatus*	EU733264.1
Dusky titi	*Callicebus moloch*	EU733265.1
Owl monkey	*Aotus trivirgatus*	FJ374685.1
Rabbit	*Oryctolagus cuniculus*	NM_001082213.1

* Nucleotide differences between the PKR sequences we used and the GenBank sequences: François leaf PKR and EU733268.1: all synonymous: T208 > C (L70L), C861 > T (I287I), G1326 > A (G442G), T1446C (H482H). Baboon PKR and XM_009184002.4: one synonymous A465G (Q155Q) and four non-synonymous G1489A (E497K), A1498C (K500Q), G1511A (G504T), and A1558G (K520E).

## Data Availability

All data necessary for the interpretation of the findings presented in this work are contained within the manuscript figures.

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
