# Peer review of "Tanapox Virus and Yaba Monkey Tumor Virus K3 Orthologs Inhibit Primate Protein Kinase R in a Species-Specific Fashion"

_viruses, 2024, doi:10.3390/v16071095_

Round 1

Reviewer 1 Report

Comments and Suggestions for Authors

This submission by Megawati et al continues the investigation of the role of K3 orthologs as a determinant of host tropism during poxvirus infection.  This study specifically focuses on orthologs from TPV and YMTV using well-established bioassays and infection studies.  With only a few exceptions mentioned below, the experiments are rigorous, methodology is fully described, and the conclusions logical and nicely integrated with historical precedents in this field of investigation.  Overall, this study clearly reinforces and extends the body of evidence that K3 orthologs are an important host-virus interaction and will be of considerable interest to the field.

Comments:

1.      Figure 2A is somewhat confusing.  It would appear that human PKR is much less repressive than other primate PKRs, is that correct?  If it is correct, then the statement “All PKRs showed strong and comparable inhibition of luciferase activity” would seem to be inaccurate. If it is not correct, then perhaps the data and normalization process need to be explained more clearly.

2.      The conclusions reached in Section 3.2 seem overly generalized and vague based on the data shown in Figure 4B.  Some acknowledgements of the role of the N-terminus, the modest fold differences in the data, and statistical differences seems warranted.  

Author Response

This submission by Megawati et al continues the investigation of the role of K3 orthologs as a determinant of host tropism during poxvirus infection.  This study specifically focuses on orthologs from TPV and YMTV using well-established bioassays and infection studies.  With only a few exceptions mentioned below, the experiments are rigorous, methodology is fully described, and the conclusions logical and nicely integrated with historical precedents in this field of investigation.  Overall, this study clearly reinforces and extends the body of evidence that K3 orthologs are an important host-virus interaction and will be of considerable interest to the field.

Comments:

  1. Figure 2A is somewhat confusing.  It would appear that human PKR is much less repressive than other primate PKRs, is that correct?  If it is correct, then the statement “All PKRs showed strong and comparable inhibition of luciferase activity” would seem to be inaccurate. If it is not correct, then perhaps the data and normalization process need to be explained more clearly.

Response: Due to an editing error the bars were mislabeled. The first bar represents the empty vector control, as correctly stated in the figure legend. Also, two bars were inadvertently labeled as Dusky titi. We corrected the labeling for this figure in this revision. Compared to the empty vector control, all PKRs showed strong and comparable inhibition of luciferase activity. We apologize for this mistake and thank both reviewers for pointing this out.

  1. The conclusions reached in Section 3.2 seem overly generalized and vague based on the data shown in Figure 4B.  Some acknowledgments of the role of the N-terminus, the modest fold differences in the data, and statistical differences seems warranted.  

Response: As suggested, we supported our conclusion that the C-terminus is important for the observed differential PKR inhibition by statistical analysis and included p-values obtained with the unpaired t-test as implemented in GraphPad to figure 4B and modified the figure legend accordingly. We agree with the reviewer that the N-terminus might have some contribution to the differential PKR inhibition, and therefore added a statement that the “we cannot exclude a contribution of the N-terminus.” (Line 321-323).

“These results indicate that the C-terminus of TPV 012 and YMTV 012 is important for the observed differential PKR inhibition, although we cannot exclude a contribution of the N-terminus.”

Reviewer 2 Report

Comments and Suggestions for Authors

These data indicate that homologues of an important immunomodulatory protein encoded by Yatapoxviruses have different inhibitory activity in different species. Similar findings have been published for other genera of poxviruses.

The abstract and introduction are well written.

Materials and methods also are well written. Writing brief explanations for why different moi's and timepoints were used for each method would be beneficial.

Results:

The sentence in lines 229-232 describes some of the data presented in Figure 1B. Additional sentences about the other comparisons could be added and that Figure could be eliminated. Alternatively, the sentence could be shortened and the Figure could be retained. (I, personally, like the figure).

The slight cross-reactivity of the anti-VACV K3 antibody with YMTV 012 is interesting. It would be nice to include a sentence or two about why you think that protein is detected by that antibody in the results or the discussion.

Figure 2A is a little surprising. Is there a reason that the luciferase assay in HeLa PKRko cells is so much more responsive to human PKR than other primate PKRs? If this is real, it would be nice to discuss this finding a bit more. Maybe the labels are incorrect though? Why are there two bars for Dusky titi PKR? Also, the letters in the labels for the species in Figure 2 are poorly spaced apart.

Figure 5B. Why were these cells chosen for this figure? Consider adding plaque assay images of the viruses in Hela PKRko cells too. It would be great to have at least one new world monkey cell line represented. The differences between BSC40 and Pro1190 (both from AGM) are a bit concerning. Do you know if PKR activation or IFN expression in response to infection differs in these two cell lines? Discussion of known differences in innate immune responses seen in different cell lines from the same species is recommended. I probably should know this, but why does vP872 form larger plaques in RK13+E3L+K3L than VC-R4? Does it have other mutations in addition to deletion of K3L?

Figure 6. It is not completely necessary, but determining viral titers of recombinant viruses that contain the chimeric 012 proteins grown in these different cells would be an interesting addition to this paper.

The discussion is well written but might benefit by addition of a few items/queries mentioned above.

Author Response

Comment: These data indicate that homologues of an important immunomodulatory protein encoded by Yatapoxviruses have different inhibitory activity in different species. Similar findings have been published for other genera of poxviruses.

The abstract and introduction are well written.

Comment: Materials and methods also are well written. Writing brief explanations for why different moi's and timepoints were used for each method would be beneficial.

Response: We added brief explanations for the different MOIs and timepoints in the method section.

We chose a MOI of 0.01 for the virus replication assays, “to assess replication in multiple cycle replication assays”. (Line 155)

For the western blots of K3 orthologs, we used “a MOI of 3 to ensure that the vast majority of cells were infected”. (Line s193-194).

For DNA extractions, we used a MOI of 0.1 as previously described [36]. (Line 165)

Comment: Results: The sentence in lines 229-232 describes some of the data presented in Figure 1B. Additional sentences about the other comparisons could be added and that Figure could be eliminated. Alternatively, the sentence could be shortened and the Figure could be retained. (I, personally, like the figure).

Response: We shortened the sentence for clarity (line 232-234) and kept the figure.

Comment:

The slight cross-reactivity of the anti-VACV K3 antibody with YMTV 012 is interesting. It would be nice to include a sentence or two about why you think that protein is detected by that antibody in the results or the discussion.

Response: We added a potential explanation for this “…, which could be explained by shared epitopes.” (Line 242).

Comment: Figure 2A is a little surprising. Is there a reason that the luciferase assay in HeLa PKRko cells is so much more responsive to human PKR than other primate PKRs? If this is real, it would be nice to discuss this finding a bit more. Maybe the labels are incorrect though? Why are there two bars for Dusky titi PKR? Also, the letters in the labels for the species in Figure 2 are poorly spaced apart.

Response: As described in the response to reviewer 1 above, this was due to an editing error. We corrected this in the revision and apologize for this mistake.

Comment: Figure 5B. Why were these cells chosen for this figure? Consider adding plaque assay images of the viruses in Hela PKRko cells too. It would be great to have at least one new world monkey cell line represented. The differences between BSC40 and Pro1190 (both from AGM) are a bit concerning. Do you know if PKR activation or IFN expression in response to infection differs in these two cell lines? Discussion of known differences in innate immune responses seen in different cell lines from the same species is recommended. I probably should know this, but why does vP872 form larger plaques in RK13+E3L+K3L than VC-R4? Does it have other mutations in addition to deletion of K3L?

Response: These cells represent several primate-derived cell lines that were available in the lab. We agree that inclusion of a new world monkey cell line might have been a nice addition; however, we do not have such a cell line available. We included RK13+E3L+K3L cells because both wild type VACV and VACV strains missing PKR inhibitors form easily recognizable plaques in these cells. The observation that VC-R4 formed smaller plaques than vP872 can be explained by incomplete PKR inhibition by the ectopically expressed E3 and K3, and is consistent with slightly less efficient replication of VC-R4 than vP872 in this cell line (Hand et al., PMID 25734776). We added the following sentence to the discussion: “VC-R4 formed smaller plaques than vP872, which can be explained by incomplete PKR inhibition by the ectopically expressed E3 and K3, and is consistent with slightly less efficient replication of VC-R4 than vP872 in this cell line [46].” (Lines 466 - 469)

To address the comment about the African Green monkey cell lines, we added the following sentence: “Although both cell lines are derived from African green monkeys, BSC-40 are immortalized epithelial cells and PRO1190 are primary fibroblasts. Differential susceptibility to various VACV recombinants by these two cell lines is well established, and may in part be due to three non-synonymous variants in PRO1190 PKR relative to BSC-40 PKR, although other host proteins, e.g. RNase L, have been implicated in some studies, suggesting this phenotype is likely multifactorial [47-49].” (Lines 478 - 484)

Comment: Figure 6. It is not completely necessary, but determining viral titers of recombinant viruses that contain the chimeric 012 proteins grown in these different cells would be an interesting addition to this paper.

Response: Although we agree that analyzing replication of recombinant viruses that contain chimeric 012 proteins would be interesting, we do not think it would add substantial new information and would add a minimum of 6 months of additional work and use substantial resources. We therefore opt not to undertake this experiment.

Comment: The discussion is well written but might benefit by addition of a few items/queries mentioned above.

Response: We addressed this above and made according modifications.
